# Multi-Layered QCA Content-Addressable Memory Cell Using Low-Power Electronic Interaction for AI-Based Data Learning and Retrieval in Quantum Computing Environment

**DOI:** 10.3390/s23010019

**Published:** 2022-12-20

**Authors:** Jun-Cheol Jeon, Amjad Almatrood, Hyun-Il Kim

**Affiliations:** 1Department of Convergence Science, Kongju National University, Gongju 32588, Republic of Korea; 2Department of Electrical Engineering, College of Engineering, Jouf University, Sakaka 72388, Saudi Arabia

**Keywords:** quantum computing, nanotechnology, artificial intelligent learning, quantum-dot cellular automata, content addressable memory, low-power QCA circuits

## Abstract

In this study, we propose a quantum structure of an associative memory cell for effective data learning based on artificial intelligence. For effective learning of related data, content-based retrieval and storage rather than memory address is essential. A content-addressable memory (CAM), which is an efficient memory cell structure for this purpose, in a quantum computing environment, is designed based on quantum-dot cellular automata (QCA). A CAM cell is composed of a memory unit that stores information, a match unit that performs a search, and a structure, using an XOR gate or an XNOR gate in the match unit, that shows good performance. In this study, we designed an XNOR gate with a multilayer structure based on electron interactions and proposed a QCA-based CAM cell using it. The area and time efficiency are verified through a simulation using QCADesigner, and the quantum cost of the proposed XOR gate and CAM cell were reduced by at least 70% and 15%, respectively, when compared to the latest research. In addition, we physically proved the potential energy owing to the interaction between the electrons inside the QCA cell. We also proposed an additional CAM circuit targeting the reduction in energy dissipation that overcomes the best available designs. The simulation and calculation of power dissipation are performed by QCADesigner-E and it is confirmed that more than 27% is reduced.

## 1. Introduction

Unlike RAM, in which data are transferred according to memory addresses, content-addressable memory (CAM) is an associative memory that returns the location of a search term or related data by searching the entire memory space when a user provides a search term [1]. While RAM performs sequential memory operations that depend on addresses, CAM operates in parallel based on all the data stored in the memory and the content to be retrieved. Therefore, it is a special memory used in search applications requiring very high speed, and is effectively used in big data, AI-based data learning, and neural networks [2,3].

Recently, as data capacity and the demand for high-speed search operations have increased, systems that require AI-based data learning, such as machine learning and deep learning are also rapidly increasing, the essential memory structure in these systems is CAM. CAM is an essential memory form used in various fields, such as machine learning, information expression, signal processing, and pattern recognition [4]. Therefore, it is very valuable to be designed in a quantum computing environment. However, it has a slightly more complicated structure than existing memory structures, such as RAM and ROM; therefore, a sensitive approach for efficient circuit design is required.

A complementary metal-oxide semiconductor (CMOS), a modern circuit design technology, has several limitations, such as current leakage due to quantum tunneling of the circuit, high-heat generation, and power consumption [5]. Quantum-dot cellular automata (QCA) is attracting attention as a next-generation technology to replace CMOS owing to several advantages, such as the size of the circuit as well as the problems mentioned above. Various quantum simulations are being tested in an upcoming quantum computing environment, and QCA is one of the most popular quantum modeling techniques [6,7]. Based on QCA, studies are being conducted to design various digital circuits as quantum circuits depending on the presence or absence of memory. This includes combinational circuits, such as XOR gates [8,9,10,11,12,13,14,15], multiplexers [16,17,18,19], adder/subtractors [20,21,22,23], multipliers [24,25], divider [26,27,28], and sequential circuits, such as D flip-flops [29,30,31,32], shift registers [33,34,35,36], RAM [37,38,39,40,41,42,43], and CAM [44,45,46,47].

Unlike RAM, which searches for information using an address, CAM is a storage device that searches for information using internally stored information. A CAM cell consists of a memory unit that stores information and a match unit that determines whether the search is successful [44]. The circuit of match unit of conventional QCA CAM cells is designed using a 5-input minority [45], 5-input majority [46], and an XOR gate [47]. However, in this study, the most structurally concise XNOR gate is used.

The existing QCA XOR gates are mostly designed based on majority vote gates, requiring a large number of cells, area, and relatively wasting space. To solve these problems, circuits using an electron interaction-based design, a method of designing a gate by inducing the operation of the circuit based on the interaction between electrons inside the QCA cell have been implemented [13,14,15]. However, the existing XOR gates are designed with a planar structure, and thus have poor space efficiency. Therefore, in this paper, we propose a multi-layered XNOR gate based on electron interaction and to design the matching unit of the CAM cell using it. The contributions of this study can be summarized as follows:An XOR/XNOR gate with a multi-layer structure based on the electron interactions is proposed;Based on the proposed XNOR gate, a CAM cell is designed and expanded to a 1 × 2 CAM to check the modularity and expandability;A memory cell targeting the optimization of area and time complexity is proposed and verified through circuit simulation;The accuracy of the designed cell is mathematically verified using physical proof;An additional circuit targeting the optimization of power consumption is proposed;An associative memory structure that can be efficiently used for data learning and retrieval based on artificial intelligence, such as neural networks, machine learning, and deep learning is designed.

The remainder of the paper is organized as follows. In Section 2, the basic background of QCA and the previously proposed XOR gates and CAM cells are discussed. Section 3 describes the proposed QCA XOR/XNOR gate and the QCA CAM cell and shows a 1 × 2 CAM circuit designed for expansion. We show physical proof of the XNOR gate proposed in Section 4 and compare and analyze the performance of area and latency with the previously proposed circuits along with the power analysis and simulation results of the proposed circuits. Finally, Section 5 summarizes the study and provides conclusions.

## 2. Related Works

In this section, the basic QCA is explained and the existing QCA XOR gates and QCA CAM cell are discussed.

### 2.1. Background of QCA

QCA uses a quantum cell as a basic element and forms a circuit by arranging adjacent quantum cells. Each quantum cell has four quantum dots that can accommodate electrons, and each cell has two electrons arranged, such that the distance between them is long owing to the Coulomb repulsion, which is a repulsive force between electrons, which are arranged diagonally. The quantum cell has two different polarizations states, which are +1 and −1 polarization as shown in Figure 1a,b, respectively, and can be mapped to 1 and 0 in binary logic. When the polarization of a specific cell is determined in the QCA circuit, the polarization of adjacent cells is also determined by Coulomb repulsion, and thus a signal is transmitted [6,7].

In the QCA environment, OR and AND operations can be implemented using majority gates. The majority gate receives three input values and has one output. Figure 2 shows an OR gate and an AND gate implemented using a majority gate that receives three inputs. As shown in Figure 2a,b, a 2-input OR gate or a 2-input AND gate can be implemented by fixing one input to 1 or 0, respectively [8,9,10,11,12,13,14,15].

The QCA circuit can be configured using multiple layers. Figure 3 shows a multilayered circuit in which a signal propagates from a quantum cell in the first layer to that in the second layer. As shown in Figure 3a, if the quantum cell of the second layer is located directly above the first layer, the signal is reversed and transmitted from Layer1 to Layer2, and the quantum cells of the two layers have opposite polarizations. On the other side, when the quantum cells of the first and second layers are positioned diagonally, the signal remains as it is and propagates normally, and the quantum cells of Layer1 and Layer2 have the same polarization as shown in Figure 3b. Figure 4a,b show the multi-layered AND/OR gates, respectively.

The QCA cell has four clock states as shown in Figure 5. These states are as follows: Switch is a process in which the potential of the cell gradually rises and becomes stronger, Hold is a state where the potential between quantum dots is maintained high and stabilized, Release is a process in which the potential is gradually decreased and weakened, and Relax is a state in which the polarization has disappeared. In QCA, these four states are used as pipeline clocks, and each pipeline clock has a phase delay of 90° from the previous clock [6,7].

### 2.2. Conventional QCA XOR Gates

Figure 6 shows the previously proposed QCA XOR gates. The existing QCA XOR gates are classified into circuits designed based on majority gates and circuits designed using electronic interactions. The circuits in Figure 6a,b are designed based on the majority gate. Here, A and B are the input values, and OUT is the output value. The fixed values −1.00 and 1.00 represent the binary values 0 and 1, respectively. Figure 6a is proposed by Mustafa et al. using four 3-input majority gates and one robust inverter. In Figure 6b, an XOR gate is designed using a 3-input majority gate, a weak inverter, and a 5-input majority gate [11,12]. Figure 6c is proposed in [13] which used a NAND gate designed by grafting the principle of electronic interaction of the majority gate and using it to develop an XOR gate with an excellent modularity and expandability. Figure 6d shows a gate proposed in [14] that performs the XOR operation with a minimum number of cells and a smaller area using electron interaction according to the cell arrangement.

### 2.3. Conventional QCA CAM Cells

Figure 7 shows a logic diagram of a 1-bit basic building block of the CAM. As shown in the figure, a CAM cell consists of a memory unit and matching unit. The memory unit stores and reads information according to the input, and the match unit checks whether the information stored in the memory matches the content to be searched for.

Table 1 presents the truth table of the memory unit. R/W that judges read and write, input, and output are displayed as I and O, and the value stored in the memory is defined as S. It is assumed that S(t − 1) is the previously stored value, S(t) is the current input value, and the input R/W determines the read and write operations of the memory unit. If R/W = 0, a write operation is performed, and the value of input I is stored in the memory, such that the equation S(t) = I is established. When R/W = 1, a read operation is performed and there is no change in the information stored in the memory unit, and the equation S(t) = S(t–1) is established. At this point, this value becomes the output value (O).

The match unit of the CAM cell serves to verify whether the argument value (A), which is the searched content, and output cell (S), which is the content stored inside the CAM cell, match. Table 2 presents the truth table of the matching unit of the CAM cell. If the key signal (K) = 0, the match value (M) = 1 regardless of the stored information, and if K = 1, the output M = 1 only when the search value A and the internally stored value S match.

Figure 8 shows the proposed QCA CAM cell circuit. Except for Figure 8d, all of the existing circuits are designed based on the logic diagram shown in Figure 7. Figure 8a is a QCA CAM cell proposed by Sardinha et al. The match unit is designed based on 3-input majority gates, and the circuit is designed in a multi-layer structure to realize the intersection of wirings [44]. Although this circuit is developed based on the fundamental units in a QCA environment, the overall circuit size is large and the delay time is long, which results in less performance efficiency. An improved design shown in Figure 8b is proposed by Heikalabad et al. based on 3-input majority gates and a 5-input minority gate [45]. Figure 8c shows another design proposed by Khosroshahy et al. In this design, the match unit is constructed using 3-input majority gates and a 5-input majority gate [46]. Finally, Figure 8d is proposed by Sadoghifar et al., and the match unit is designed using a 2-input XOR gate and a 3-input majority gate. The area and delay time in this design are considerably reduced by reducing the circuit wiring and using an optimized XOR gate developed based on electronic interaction [47].

## 3. Proposed CAM Cell

Although the required area of the QCA XOR gate has been improved by designing it using electron interaction, it is designed as a single-layer structure, which results in a spatial limitation. Therefore, this paper proposes a QCA XOR gate with an improved space efficiency using a multi-layer structure and an electron interaction design technique. Figure 9 shows the proposed XOR/XNOR circuit designed with a two-layer structure and based on the electronic interaction to increase the space efficiency compared to the existing circuits. In addition, the proposed circuit can implement an XNOR gate without using an inverter by changing the polarization of the fixed cells in the second layer from −1 to 1 as shown in Figure 9c.

Figure 10 shows the logic diagram of the CAM cell used in this study. The circuit that determines the output value M in the match unit of Figure 7 is arranged as in Equation (1) and can be designed using a 2-input XNOR and a 2-input OR gate.
(1)$$\begin{array}{ll} M & ={\left(\left(A+S\right)\cdot \left(A^{\prime }+S^{\prime }\right)\cdot K\right)}^{\prime }\\ & =A^{\prime }\cdot S^{\prime }+A\cdot S+K^{\prime }\\ & =A⊙S+K^{\prime } \end{array}$$

Figure 11 shows each layer of the QCA CAM cell separately based on the logic diagram of Figure 10. The position of the cells can be identified using the index written on the row and column of each figure. The AND/OR gates used in the circuit are multi-layer structures given in Figure 4. The XNOR gate suggested in Figure 9 is used in the match unit. For signal stability, two redundant cells are inserted into Layer2 (b,10) and Layer3 (c,11). 

Figure 12 shows the circuit expanded to a 1 × 2 CAM using the proposed QCA CAM cell. In the two CAM cells, the operation of the memory unit is controlled according to the same R/W input, and each cell of the different inputs K, I, and A, has a different input. M0 and M1, the outputs of the match unit of each cell, are AND operation performed. When both cells of the 1 × 2 CAM are successfully searched, the resulting value of Match output is 1.

## 4. Physical Proof and Performance Analysis

### 4.1. Physical Proof of the Proposed XNOR Gate

Physical proof is used to mathematically verify the operation of a circuit in QCA. It is a proof method used to assume the polarization of a cell, when its polarization is not determined as +1 or −1 and to determine the polarization of the cell through potential energy in each state [48]. First, the polarization of the targeted cell for which the state to be obtained is assumed. The potential energy interacting with the electrons of the cell whose neighboring state is then determined and calculated. Then the sum of potential energies is calculated. This is implemented when the polarization of the target cell is either +1 or −1. By comparing the values of potential energy, the polarization with the lower value is determined as the polarization of the targeted cell.

Equation (2) is a formula for calculating the potential energy owing to the interaction between electrons.  U is the potential energy, *k* is Coulomb’s constant, *q*_1_ and *q*_2_ are the sizes of the electric charges of each electron, and *r* is the distance between the two electrons. At this time, *k*, *q*_1_, and *q*_2_ are constants, calculating the numerator of Equation (2) as given in Equation (3). In addition, the potential energy of the electron configuration of the cell is expressed as the sum ($U_{T}$) of the potential energies of the two electrons in the cell as in Equation (4) [48]:(2)$$U=\frac{kq_{1}q_{2}}{r}$$
(3)$$kq_{1}q_{2}=9\times 10^{9}\times 1.6^{2}\times 10^{-38}=23.04\times 10^{-29}$$
(4)$$U_{T}=\overset{2}{\underset{i=1}{\sum }}U_{i}$$

Before providing physical proof, this study makes the following assumptions: First, it is assumed that all cells are square with a side length of 18 nm and a cell-to-cell distance of 2 nm. Second, it is assumed that the quantum dots of the QCA cell are located at the vertices of the cell, and the electrons are also located at the center of each quantum dot. Third, the maximum electron-to-electron interaction distance is assumed to be 80 nm.

Figure 13 shows the number of electrons and cell locations assumed for the physical proof of the proposed XNOR gate. There are four cells with unknown polarizations that are numbered from 1 to 4 and it is required to find the polarization corresponding to each cell. Figure 13a,b are the first-layer structures of the proposed XNOR gate, and Figure 13c is the second-layer structure.

The positions of electrons inside the input cell are assumed to be e1 to e4, and as the fixed cells have a polarization of +1, the electrons inside are located at e5 to e8, respectively. The electrons inside the cell to be obtained are denoted by *X* and *Y*, respectively, and the use of physical proof is carried out in the order of cell 1 to 4.



$$E_{e1X}=\frac{23.04\times 10^{-29}}{2.00\times 10^{-9}}=1.152\times 10^{-19}\;E_{e2X}=\frac{23.04\times 10^{-29}}{26.91\times 10^{-9}}=8.563\times 10^{-21}\;E_{e3X}=\frac{23.04\times 10^{-29}}{20.00\times 10^{-9}}=1.152\times 10^{-20}\;E_{e4X}=\frac{23.04\times 10^{-29}}{42.05\times 10^{-9}}=5.480\times 10^{-21}\;E_{e5X}=\frac{23.04\times 10^{-29}}{21.36\times 10^{-9}}=1.079\times 10^{-20}\;E_{e6X}=\frac{23.04\times 10^{-29}}{21.36\times 10^{-9}}=1.079\times 10^{-20}\;E_{e7X}=\frac{23.04\times 10^{-29}}{41.67\times 10^{-9}}=5.529\times 10^{-21}\;E_{e8X}=\frac{23.04\times 10^{-29}}{61.81\times 10^{-9}}=3.728\times 10^{-21}$$



$$E_{e1Y}=\frac{23.04\times 10^{-29}}{26.91\times 10^{-9}}=8.563\times 10^{-21}\;E_{e2Y}=\frac{23.04\times 10^{-29}}{38.00\times 10^{-9}}=6.063\times 10^{-21}\;E_{e3Y}=\frac{23.04\times 10^{-29}}{18.11\times 10^{-9}}=1.272\times 10^{-20}\;E_{e4Y}=\frac{23.04\times 10^{-29}}{20.00\times 10^{-9}}=1.152\times 10^{-20}\;E_{e5Y}=\frac{23.04\times 10^{-29}}{21.36\times 10^{-9}}=1.079\times 10^{-20}\;E_{e6Y}=\frac{23.04\times 10^{-29}}{21.36\times 10^{-9}}=1.079\times 10^{-20}\;E_{e7Y}=\frac{23.04\times 10^{-29}}{31.88\times 10^{-9}}=7.227\times 10^{-21}\;E_{e8Y}=\frac{23.04\times 10^{-29}}{56.36\times 10^{-9}}=4.088\times 10^{-21}$$



First, the potential energy of cell 1 = −1 is analyzed. In (a) and (b) of Figure 13, the values of the potential energy are calculated with each adjacent electron in *X* and *Y* from e1 to e8 as given in (5) and (6). $U_{X}$ and $U_{Y}$ denote the potential energies of *X* and *Y*, respectively: (5)$$U_{X}=\overset{\;8}{\underset{i=1}{\sum }}E_{eiX}=1.716\times 10^{-19}$$
(6)$$U_{Y}=\overset{\;8}{\underset{i=1}{\sum }}E_{eiY}=7.176\times 10^{-20}$$

For example, $E_{e1X}$ refers to the potential energy of e1 and *X* when interacting with each other. Therefore, the potential energy when cell 1 has a polarization of −1 is equal to (7) because it is the sum of $U_{X}$ and $U_{Y}$:(7)$$U_{T}=U_{X}+U_{Y}=1.716\times 10^{-19}+7.176\times 10^{-20}=2.433\times 10^{-19}$$

Similarly, in (b) and (c) of Figure 13, the potential energy when cell 1 has +1 polarization, the result can be obtained by Equation (8):(8)$$U_{T}=U_{X}+U_{Y}=8.139\times 10^{-20}+1.773\times 10^{-19}=2.587\times 10^{-19}$$

A comparison of Equations (7) and (8) shows that the potential energy when the polarization (P) of cell 1 = −1, is smaller than the potential energy when it is +1; therefore, the polarization of cell 1 is determined to be −1. After that, the state of cell 1, in which the polarization is determined, is added. Physical proof of cell 2 is also performed and the result is shown in Equation (9):(9)$$\;\;\;\;\;\;\;\;\;\;\;\;\;\;\;\;P=-1:\;\;U_{T}=U_{X}+U_{Y}=1.041\times 10^{-19}+7.074\times 10^{-20}=1.749\times 10^{-19}\;P=+1:\;\;U_{T}=U_{X}+U_{Y}=1.934\times 10^{-19}+8.183\times 10^{-20}=2.752\times 10^{-19}$$

As shown in Equation (9), when the polarization of cell 2 is −1, the potential energy is lower, and thus the polarization of cell 2 is determined to be −1. In the same way, by using physical proof and adding the polarization of the cells determined in the previous step, the polarization of cell 3 can be calculated. The polarization of cell 4 can also be obtained based on the polarization of cell 3. By analyzing the polarization of cells 1, 2, 3, and 4 determined by using physical proof, cell 1 has the result of an OR operation of A and B, and cells 2, 3, and 4 have the result of an XNOR operation of A and B.

### 4.2. Simulation and Analysis

All circuits introduced in this study are simulated using QCADesigner 2.0.3 [49]. Bistable approximation and a coherence vector simulation engine are used and the factors are set as listed in Table 3. Figure 14 shows the simulation results for the proposed XOR/XNOR gate. From Figure 14a, it is observed that the operation of XOR gate and the operation of XNOR gate in Figure 14b are normally performed.

Figure 15 shows the simulation result of the CAM cell. This shows that the proposed circuit works well and has sufficient signal stability.

In the tables below, we compare and analyze the performance of existing circuits and the circuit proposed in this study. The total number of cells and space used for a circuit design are expressed as cell count and area, respectively. The time in clock cycle units used from the input cell to the final output cell is also compared and expressed as latency. At this time, one clock cycle consists of four clocks. In addition, cost is compared and calculated by the product of area and the square of latency as shown in Equation (10) [50]. When comparing the performance of circuit designs, latency is often used as an arithmetic expression that further emphasizes its importance. For cost, the decimal values are rounded up to negligible values. Table 4 compares the performance of the proposed XOR gate and conventional circuits:(10)$$Cost=Area\times Latency^{2}$$

The circuit with the best performance among the previously proposed gates is the XOR gate using electron interaction as proposed by Chabi et al., which takes a single unit of clock time and requires only a space of 11,564 nm^2^ [14]. The proposed XOR gate has the same required latency as the existing XOR gate with the best performance and reduces the required area by more than 70%, by using an electron interaction-based multi-layer structure.

Table 5 compares the performance of the proposed QCA circuit of the CAM cell and the circuits previously designed with the best performance. Compared to the existing circuits, the proposed CAM cell reduced the required area by at least 15% and up to 91% and shortened the required latency by up to 74%. Consequently, a large performance improvement is obtained at an overall cost.

Table 6 confirms the modularity and expandability of the circuit by extending the 1-bit CAM cell compared in Table 5 to the 2-bit, and confirms the area and clock consumed for the additional wiring required for the connection between the CAM cells. For expansion, all circuits are designed with the same three-layer structure and the connection part is designed in a manner similar to the expansion circuit shown in Figure 12. As given in Table 5, the proposed CAM performed 5% better than the circuit proposed in [47], but as a result of expanding the circuit to 1 × 2 CAM, it showed a superior performance by more than 15%. This shows the high modularity and expandability of the proposed CAM cell and shows that it is more cost-effective as the circuit expanded.

Table 7 shows the energy dissipation values estimated by QCADesigner-E (rounded up to the second decimal place) [51]. The energy loss rate is determined by the size of the circuit, the density of cells, the number of inputs/outputs and the number of crossovers of the wiring and other design factors. Therefore, the multi-layer structure has a relatively large energy loss rate. The circuit proposed in Figure 11 is designed with an optimization priority given to area and latency. However, it does not significantly contribute in reducing the energy loss due to the multi-layered structure and redundant cells. Therefore, the multi-layered K input cell is moved from Layer2 to Layer1 and the redundant cells are eliminated in order to reduce the energy dissipation of the proposed circuit as shown in Figure 16. Furthermore, one additional cell is inserted before the M output cell to reduce the density of cells. This results in an energy dissipation reduction of 34% when compared to our first design. The proposed design has achieved an approximate energy dissipation reduction of 27% when compared to the best comparable available design in [45]. The circuit in [44] has only two inputs and one output, and the circuit in [47] does not have an output O. Therefore, they cannot be compared on equal terms regarding energy dissipation due to the equal comparison limitation of circuits when the number of inputs and outputs is different.

## 5. Conclusions

In this study, we designed a space-efficient XOR gate using a multi-layer structure in a QCA environment. Based on this design, a CAM cell is also proposed. The proposed XOR gate developed using electron interaction requires only a small area. By changing the state of the fixed cells in the second layer, an XNOR gate can be implemented without using an inverter. In addition, a multilayered CAM cell is designed using the proposed XOR/XNOR structure and extended to 1 × 2 CAM. The operation is verified using physical proof and simulation to demonstrate the modularity and scalability of the circuit. It also shows how to minimize the energy dissipation in the circuits designed with a multi-layer structure. By comparing the proposed designs with existing circuits, the overall performance is improved by optimizing the energy dissipation and space efficiency while maintaining the same latency, and it is confirmed that the circuit performance improved as the circuit expanded. The proposed research reduces the quantum cost by 70% and 15% when compared to the existing XOR gates and CAM cells, respectively. Moreover, it reduces energy dissipation by more than 27% when compared to the best existing design. In the energy loss comparison, it is difficult to compare circuits with different numbers of inputs and outputs on an equal basis, and there are clearly limitations in the trade-off between quantum cost and energy dissipation. Therefore, continuous research into how to overcome these issues is needed. The expanded N × N CAM circuit using the proposed circuit is expected to be used efficiently as a memory circuit for data learning and retrieval based on artificial intelligence, such as neural networks, machine learning, and deep learning in a quantum computing environment.

## Figures and Tables

**Figure 1 sensors-23-00019-f001:**
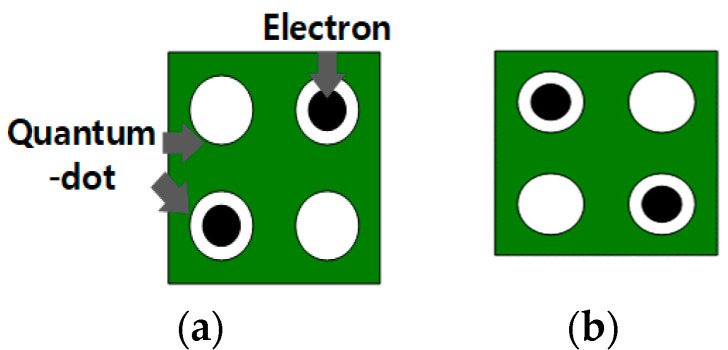
Two polarization states of QCA basic cell: (**a**) P = +1 (binary 1); (**b**) P = −1 (binary 0).

**Figure 2 sensors-23-00019-f002:**
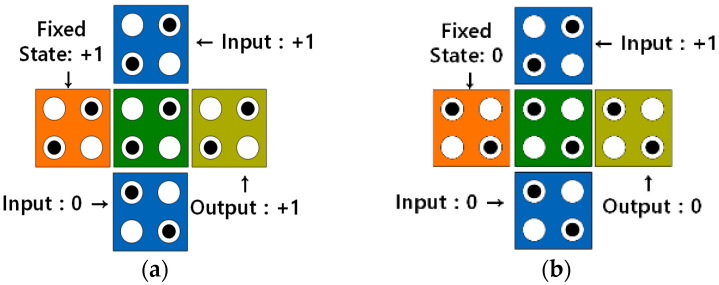
3-input majority gate: (**a**) OR gate; (**b**) AND gate.

**Figure 3 sensors-23-00019-f003:**
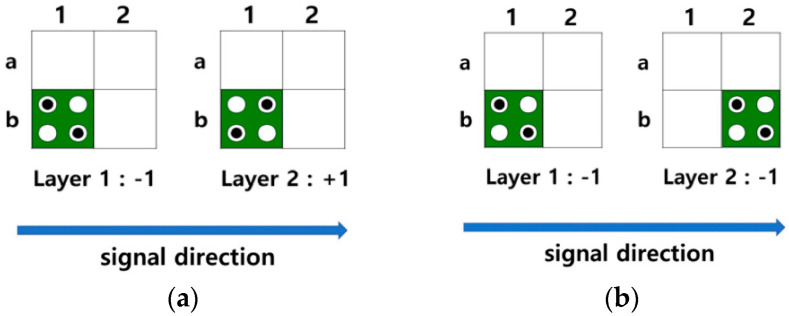
Signal transmission in multi-layer: (**a**) Grid expression of vertical transmission; (**b**) Grid expression of diagonal transmission.

**Figure 4 sensors-23-00019-f004:**
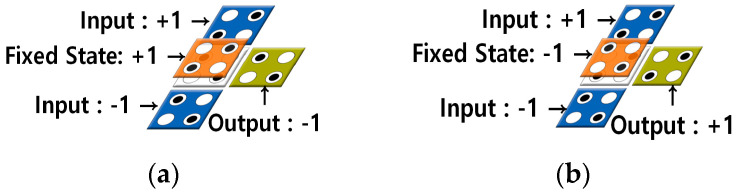
Multilayer based AND/OR gate: (**a**) AND gate, (**b**) OR gate.

**Figure 5 sensors-23-00019-f005:**
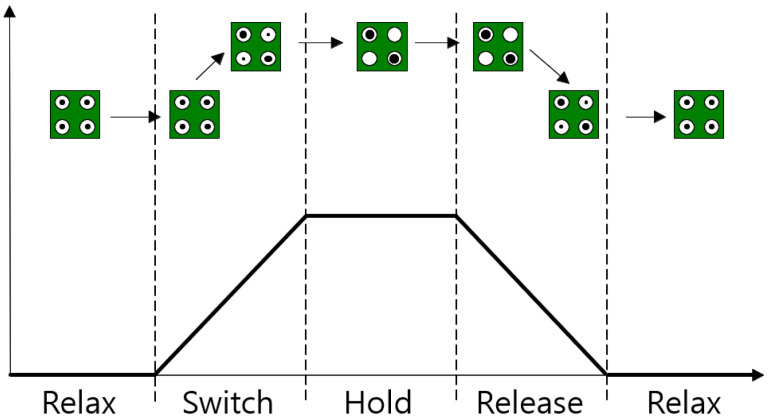
Four states of the QCA clock.

**Figure 6 sensors-23-00019-f006:**
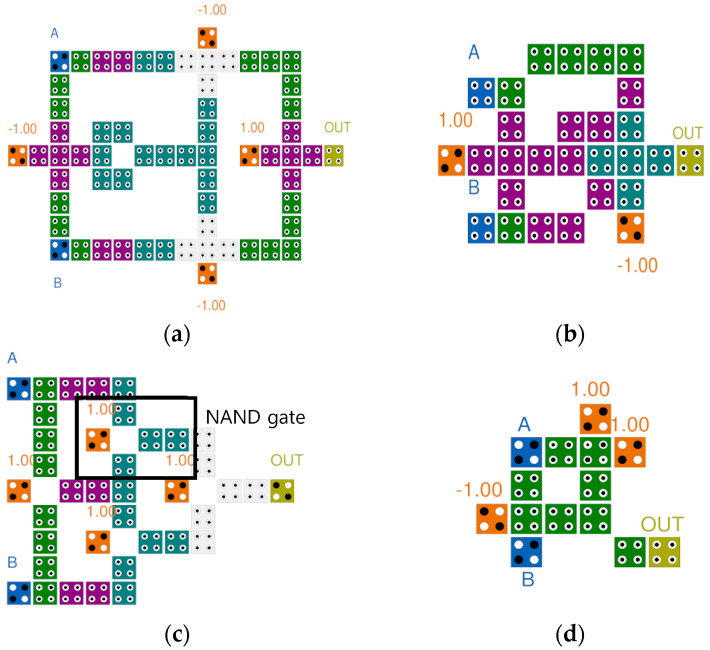
Conventional QCA XOR gates: (**a**) M. Mustafa et al.’s [11]; (**b**) M. B. Khosroshahy et al.’s [12]; (**c**) M. Poorhosseini et al.’s [13]; (**d**) A. M. Chabi et al.’s [14].

**Figure 7 sensors-23-00019-f007:**
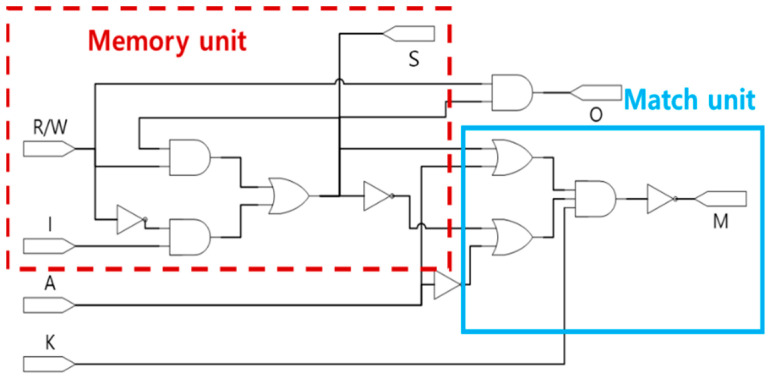
Logic diagram of CAM cell.

**Figure 8 sensors-23-00019-f008:**
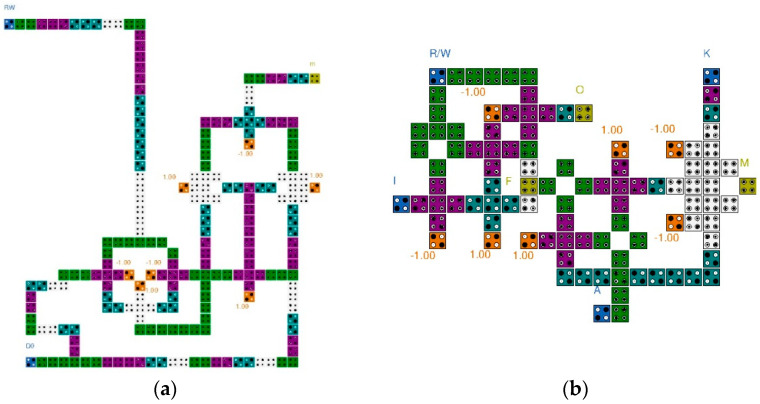
Conventional QCA CAM cells: (**a**) L. H. B. Sardinha et al.’s [44]; (**b**) S. R. Heikalabad et al.’s [45]; (**c**) M. B. Khosroshahy et al.’s [46]; (**d**) A. Sadoghifar et al.’s [47].

**Figure 9 sensors-23-00019-f009:**
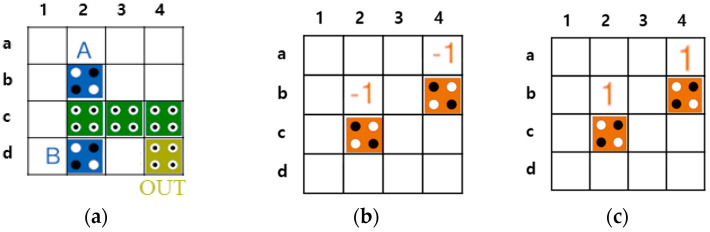
Proposed XOR/XNOR gates: (**a**) Layer 1-common structure; (**b**) Layer 2-XOR structure; (**c**) Layer 2-XNOR structure.

**Figure 10 sensors-23-00019-f010:**
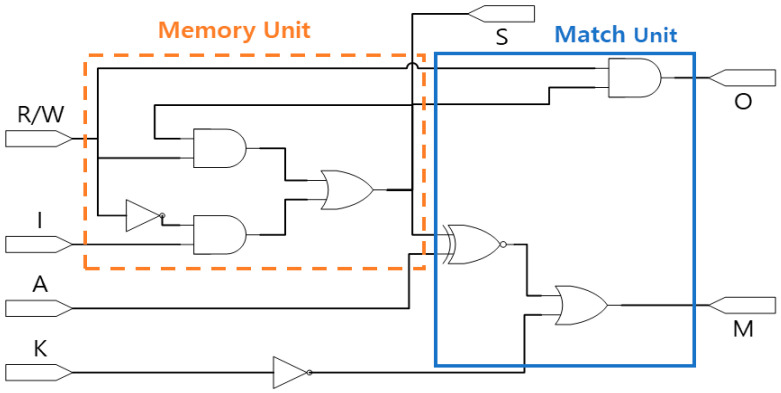
Logic diagram of XNOR-based CAM cell.

**Figure 11 sensors-23-00019-f011:**
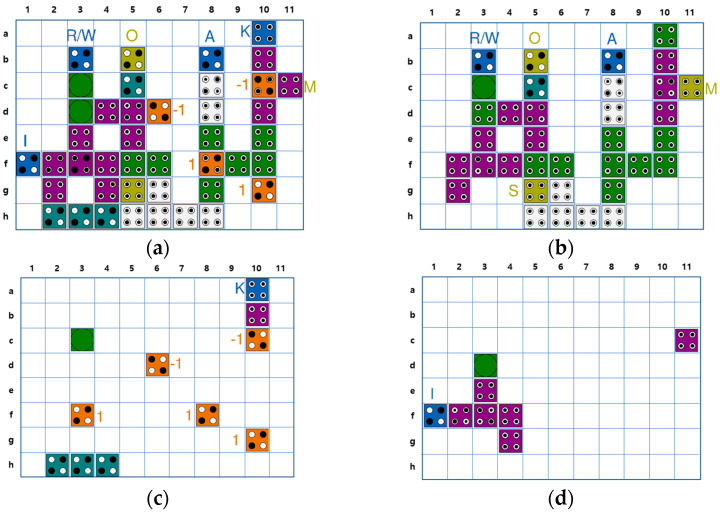
Proposed QCA CAM cell: (**a**) All layer; (**b**) Layer 1; (**c**) Layer 2; (**d**) Layer 3.

**Figure 12 sensors-23-00019-f012:**
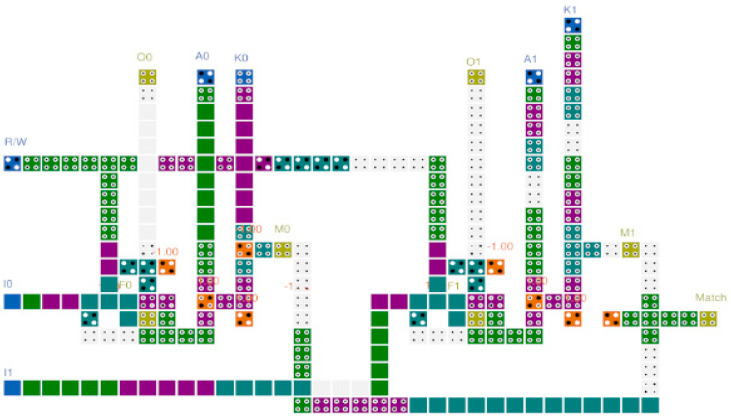
Proposed 1 × 2 CAM circuit.

**Figure 13 sensors-23-00019-f013:**
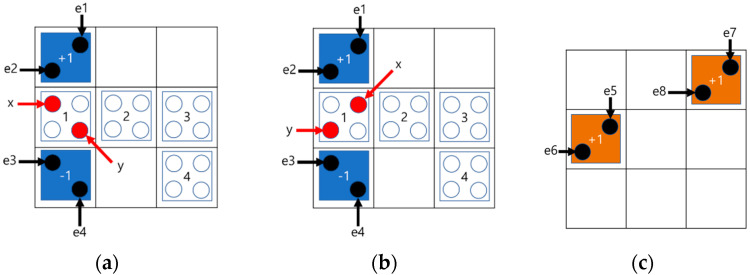
Location of electrons and cells for physical proof of the proposed XNOR gate: (**a**) Layer 1: cell 1 = −1 (**b**) Layer 1: cell 1 = +1; (**c**) Layer 2.

**Figure 14 sensors-23-00019-f014:**
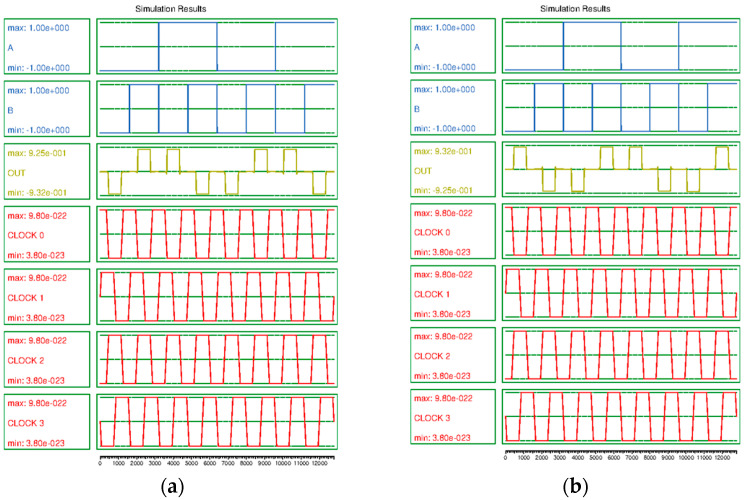
Simulation result: (**a**) XOR gate, (**b**) XNOR gate.

**Figure 15 sensors-23-00019-f015:**
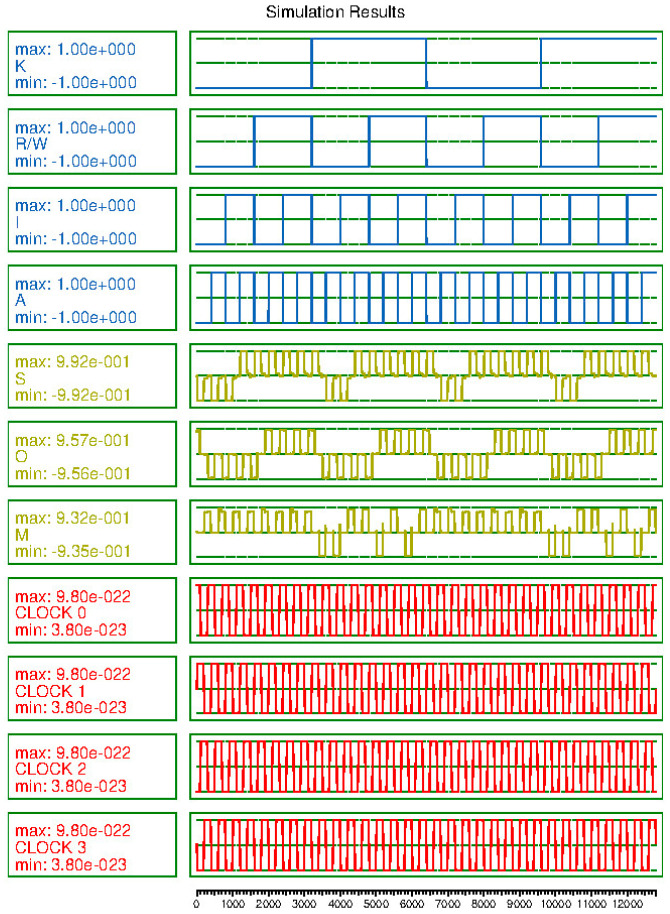
Simulation result of proposed CAM cell.

**Figure 16 sensors-23-00019-f016:**
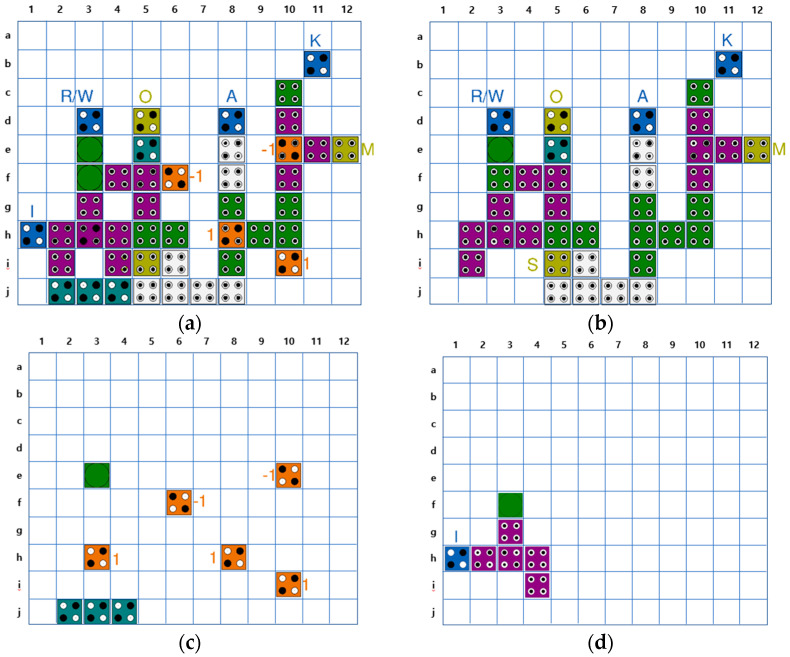
Energy Optimized QCA CAM cell: (**a**) All layer; (**b**) Layer 1; (**c**) Layer 2; (**d**) Layer 3.

**Table 1 sensors-23-00019-t001:** Truth table of CAM cell memory unit.

R/W	I	S(t − 1)	S(t)	O
0	0	X	0	0
0	1	X	1	0
1	X	0	0	0
1	X	1	1	1

**Table 2 sensors-23-00019-t002:** Truth table of CAM cell match unit.

K	A	S	M
0	X	X	1
1	0	0	1
1	0	1	0
1	1	0	0
1	1	1	1

**Table 3 sensors-23-00019-t003:** Simulation parameters.

Parameters	Bistable Approximation	Coherence Vector
Cell size	18 nm	18 nm
Dot diameter	5 nm	5 nm
Cell separation	2 nm	2 nm
Layer separation	11.5 nm	11.5 nm
Clock high	9.8 × 10^−22^ J	9.8 × 10^−22^ J
Clock low	3.8 × 10^−23^ J	3.8 × 10^−23^ J
Clock shift	0	0
Clock amplitude factor	2.0	2.0
Relative permittivity	12.9	12.9
Number of samples	12800	-
Maximum iteration per sample	100	
Convergence tolerance	1.0 × 10^−3^	-
Temperature	-	1 K
Relaxation time	-	1.0 × 10^−15^ s
Time step	-	1.0 × 10^−16^ s
Radius of effect	65 nm	80 nm

**Table 4 sensors-23-00019-t004:** Comparison of XOR gates.

Circuit	Cell Count	Area (nm^2^)	Latency (Clock Cycle)	Cost	Crossover
[11]	62	70,278	1	70,278	Coplanar
[12]	28	21,004	0.75	11,815	Coplanar
[13]	38	38,804	1	38,804	Coplanar
[14]	14	11,564	0.25	723	Coplanar
[15]	17	19,044	0.5	4761	Coplanar
Proposed	8	3364	0.25	210	Multilayer

**Table 5 sensors-23-00019-t005:** Comparison of QCA CAMs.

Circuit.	Area (nm^2^)	Latency (Clock Cycle)	Cost	Crossover
[44]	368,764	5.75	1.22 × 10^7^	Multilayer
[45]	110,644	2	4.43 × 10^5^	Coplanar
[46]	110,644	2	4.43 × 10^5^	Coplanar
[47]	40,764	1.5	9.17 × 10^4^	Coplanar
Proposed	34,444	1.5	7.75 × 10^4^	Multilayer

**Table 6 sensors-23-00019-t006:** Comparison of QCA 1 × 2 CAMs.

Circuit	Area (µm^2^)	Latency (Clock Cycle)	Cost	Crossover
[44]	0.92	7	45.08	Multilayer
[45]	0.72	3.75	10.13	Multilayer
[46]	0.71	3.75	9.98	Multilayer
[47]	0.40	3.25	4.23	Multilayer
Proposed	0.34	3.25	3.59	Multilayer

**Table 7 sensors-23-00019-t007:** Energy dissipation comparison of QCA CAM cells.

	[44]	[45]	[46]	[47]	Figure 11	Figure 16
The number of (input, output)	3 (2, 1)	7 (4, 3)	7 (4, 3)	6 (4, 2)	7 (4, 3)	7 (4, 3)
Average energy dissipatioin (meV)	4.56	3.22	3.72	1.45	3.55	2.34

## Data Availability

Data is contained within the article.

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
