# Peer review of "Multi-Layered QCA Content-Addressable Memory Cell Using Low-Power Electronic Interaction for AI-Based Data Learning and Retrieval in Quantum Computing Environment"

_sensors, 2022, doi:10.3390/s23010019_

Round 1

Reviewer 1 Report

title must be shorten and should reflect the objective of the study.

I have some recommendations for revision before the article can be formally accepted for publication. 

1. The abstract is very confusing and does not leave any impressive impression. I suggest redrafting it according to the following guidelines.

The first two lines must encompass the context of the study and the research problem, further two lines must be covered the objective of the papers with unfolding the description of the title. In the next 2 to 4 lines the methodology will be covered. Afterward, the next two lines are for result and performance. In these lines, the author must define how the results and performance are being achieved, for instance, by conducting either simulation or physical implementation. Please mention the name of the simulation or the physical method. The result statistics must be mentioned in the last two lines and either in percentage or with real-time values. 

2. The authors must give a clear procedure for the proposed solution with an algorithm (via flowcharts or pseudo codes, i.e. the flowcharts of a proposal work must be drawn), and must be supported with a figure either a block diagram of the proposed methodology or the topology in a formal style.

3. In the Literature Review section, the authors must cover the shortcomings because shortcoming or challenges in previous work is not mentioned. If the previous work is free of challenges and there is no issue then what motivated the authors to propose this study? It inculcates that authors should revise the literature review and critically highlight the problems in the previous study and compare the proposed solution and tells how the proposed solution is best fitted. 

4. In addition, please add a table and present the summary of related work, the shortcomings, and the proposed solution accordingly.

5. In the Research Methodology section, what is the difference between your proposed method and other based techniques? (II) Please compare your proposed method with other based methods in terms of time complexity.

6. Critical analysis of the finding, which is the most important part, is missing. This would help the readers to further improve the study.

7. Go for a thorough proofread of the paper to rectify several existing typos and grammatical mistakes to improve the written quality of the paper. If necessary take the help of a native English speaker to improve the language of the paper.

8. Regarding the conclusion paragraph, Please precisely describe the outcome of the study and justify the statements that are mentioned in the abstract. Further, it must contain additional points and must give a clear and more discussion about the experimental results. The main novelty and contribution of needs must be summarized and highlight the recommendations based on obtained results. These results are the hallmark for future extension therefore, please spend some more time writing the conclusion and based on the results suggest new directions.

9. While reviewing the references, I observed that cited references are outdated whereas more work already has been done in the proposed study. The cited references are neither sufficient nor suitable and therefore must extend the list and focus only on the papers from the recent 3 years. The following references may be added to supersede the outdated ones for the authors' convenience.

https://doi.org/10.26634/jwcn.8.3.17310

Author Response

  1. The abstract is very confusing and does not leave any impressive impression. I suggest redrafting it according to the following guidelines.

The first two lines must encompass the context of the study and the research problem, further two lines must be covered the objective of the papers with unfolding the description of the title. In the next 2 to 4 lines the methodology will be covered. Afterward, the next two lines are for result and performance. In these lines, the author must define how the results and performance are being achieved, for instance, by conducting either simulation or physical implementation. Please mention the name of the simulation or the physical method. The result statistics must be mentioned in the last two lines and either in percentage or with real-time values. 

  • The abstract has been modified according to the reviewer's advice on page 1

  1. The authors must give a clear procedure for the proposed solution with an algorithm (via flowcharts or pseudo codes, i.e. the flowcharts of a proposal work must be drawn), and must be supported with a figure either a block diagram of the proposed methodology or the topology in a formal style.
  • The problem-solving method of the proposed study is the design of circuits using multi-layer structures and electronic interactions. The problem-solving method is so intuitive that I do not think there is a need for algorithms or flow charts. The beginning of a major contribution is the design of XOR/XNOR gates using multilayer structures. As shown in Figure 6, it was intuitively designed using the interaction between electrons to overcome the design limitations of the existing structures, and the clear results are shown in Figure 14. Based on this, the QCA CAM cell was designed, and the result is also well shown in Figure 15. Please understand that the design of the QCA circuit is a difficult task in which, due to the instability of electrons and interactions between electrons, it is difficult to find answers through numerous design mistakes and to prove them in reverse rather than presenting clear algorithms and rules.

  1. In the Literature Review section, the authors must cover the shortcomings because shortcoming or challenges in previous work is not mentioned. If the previous work is free of challenges and there is no issue then what motivated the authors to propose this study? It inculcates that authors should revise the literature review and critically highlight the problems in the previous study and compare the proposed solution and tells how the proposed solution is best fitted.
  • In Section 1, Introduction, the problems of existing studies including CMOS, QCA CAM, and XOR gate were raised and the contribution of the proposed research was summarized. Also, in Section 2, the existing XOR gate and CAM structures are mentioned in Figures 6 and 7 and the problems of the existing structures are reviewed. In Section 3, it was mentioned that multilayer structures and electronic interactions are used to design to overcome spatial limitations. Also, as shown in Figure 10, the existing CAM cell was modified and supplemented with an XNOR-based CAM cell using equation (1), and the QCA CAM to which the multi-layered XNOR structure of Figure 9 was applied is shown in Figure 11. Comparison and analysis are presented in Section 4.

  1. In addition, please add a table and present the summary of related work, the shortcomings, and the proposed solution accordingly.
  • In introduction, the shortcomings of the existing studies were mentioned, and it was specified that the existing problems were solved by using the interaction between electrons and the multi-layered structure on page 2.

  1. In the Research Methodology section, what is the difference between your proposed method and other based techniques? (II) Please compare your proposed method with other based methods in terms of time complexity.
  • Existing studies have been designed in a single-layer structure based on majority voting gates. The proposed study minimized quantum cost by designing a multi-layered structure using electron interaction. Time complexity is clearly expressed in Tables 4 to 6 under the name of Latency in units of clock cycles.

  1. Critical analysis of the finding, which is the most important part, is missing. This would help the readers to further improve the study.
  • In the energy loss comparison of circuits, it is difficult to compare circuits with different numbers of inputs and outputs on an equal basis, and there is clearly a limit in the trade-off between quantum cost and energy dissipation on page 14.

  1. Go for a thorough proofread of the paper to rectify several existing typos and grammatical mistakes to improve the written quality of the paper. If necessary take the help of a native English speaker to improve the language of the paper.
  • This manuscript was proofread in English by a professional editing company. (Certificate attached)

  1. Regarding the conclusion paragraph, Please precisely describe the outcome of the study and justify the statements that are mentioned in the abstract. Further, it must contain additional points and must give a clear and more discussion about the experimental results. The main novelty and contribution of needs must be summarized and highlight the recommendations based on obtained results. These results are the hallmark for future extension therefore, please spend some more time writing the conclusion and based on the results suggest new directions.
  • According to the reviewer's advice, the result part has been modified.

  1. While reviewing the references, I observed that cited references are outdated whereas more work already has been done in the proposed study. The cited references are neither sufficient nor suitable and therefore must extend the list and focus only on the papers from the recent 3 years. The following references may be added to supersede the outdated ones for the authors' convenience.

https://doi.org/10.26634/jwcn.8.3.17310

Thanks for the advice, we have taken good care of the latest relevant studies.

Reviewer 2 Report

The manuscript entitled Multi-layered QCA Content-Addressable Memory Cell Using 

Low-Power Electronic Interaction for AI-Based Data Learning

and Retrieval in Quantum Computing Environment investing very good problem. The introduction defines the background and contributions of the work. The related work discussed existing efforts to the considered problem. The problem defination defined the systematic flow of the problem. The proposed algorithms and results shown that the proposed work has optimal results. However, still in paper many mistakes exist. 

1. Many symbol notations are exploited, there must be a Table where all symbols must be defined in clear way. 

2. Why simulation parameters didn't consider the all equation in the system? 

3. There should be algorithm flow about how to solve problem in different steps.

4. Paper is tricky, the must be some case studies about considered problem.

5. Finding and limitations section must be added before conclusion.

6. Time complexity is not clear of the proposed work.

7. Data type and features are missing in manuscript.

8. Results are not defined well. 

Author Response

  1. Many symbol notations are exploited, there must be a Table where all symbols must be defined in clear way. 
  • To help clear definition and understanding of symbols, definitions of symbols are added wherever symbols appear instead of a symbol table on page 4. All other symbols have been added.

  1. Why simulation parameters didn't consider the all equation in the system?
  • Most of the equations (2) to (9) in the proposed manuscript are related to physical proof, which means the correlation with the position of the electron, and only the distance was used with meaning in the simulation parameters.

  1. There should be algorithm flow about how to solve problem in different steps.
  • The problem-solving method of the proposed study is the design of circuits using multi-layer structures and electronic interactions. The problem-solving method is so intuitive that I do not think there is a need for algorithms or flow charts. The beginning of a major contribution is the design of XOR/XNOR gates using multilayer structures. As shown in Figure 6, it was intuitively designed using the interaction between electrons to overcome the design limitations of the existing structures, and the clear results are shown in Figure 14. Based on this, the QCA CAM cell was designed, and the result is also well shown in Figure 15. Please understand that the design of the QCA circuit is a difficult task in which, due to the instability of electrons and interactions between electrons, it is difficult to find answers through numerous design mistakes and to prove them in reverse rather than presenting clear algorithms and rules.

  1. Paper is tricky, the must be some case studies about considered problem.
  • In Fig. 6 and Fig. 8, examples of the previously designed XOR gate and CAM cell are shown, and related explanations are added on page 4 to 7.

  1. Finding and limitations section must be added before conclusion.
  • In the energy loss comparison of circuits, it is difficult to compare circuits with different numbers of inputs and outputs on an equal basis, and it is clarified that there is a clear limit of the trade-off between quantum cost and energy dissipation on page 14.

  1. Time complexity is not clear of the proposed work.
  • Time complexity is clearly expressed in Tables 4 to 6 under the name of Latency in units of clock cycles.

  1. Data type and features are missing in manuscript.
  • In designing the QCA circuit, it is thought that the description of the data type is not necessary.

  1. Results are not defined well. 
  • Various comparisons between the designed circuit and the existing circuits are well shown in Tables 4 to 7, and the comparison and analysis of how good the relatively good results were made or written in detail below the tables. In addition, the conclusion part was also corrected and supplemented along with the result figures.

Reviewer 3 Report

The authors have proposed a quantum structure of an associative memory cell for effective data learning based on artificial intelligence. The authors have selected a good problem and proposed a novel solution which sounds technical. The paper is well written

I recommend to accept the paper in current form.

Author Response

  1. The authors have proposed a quantum structure of an associative memory cell for effective data learning based on artificial intelligence. The authors have selected a good problem and proposed a novel solution which sounds technical. The paper is well written

I recommend to accept the paper in current form.

  • Thank you for your sincere advice.

Reviewer 4 Report

This manuscript presents a quantum structure of an associative memory cell for data learning based on artificial intelligence. The manuscript is well writtern and clear. This result could be of interest to experts in this field. The manuscript could become suitable for publication in Sensors provided that the authors satisfactorily reply to the below comments.

The authors claim that the proposed structure could be used in machine learning in a quantum computing environment. I think some cases need to be discussed.

When the authors introduce the quantum computing in the ``Introduction'' section, it may raise interest to a broader readership if some references about the development of different quantum computing systems are provided, such as

(i) Blatt, Rainer, and Christian F. Roos. Quantum simulations with trapped ions. Nature Physics 8.4 (2012): 277-284.

(ii) Bruzewicz, Colin D., et al. Trapped-ion quantum computing: Progress and challenges. Applied Physics Reviews 6.2 (2019): 021314.

(iii) Huang, He-Liang, et al. Superconducting quantum computing: a review. Science China Information Sciences 63, 180501 (2020)

And also some references about quantum machine learning, such as

i)Seth Lloyd and Christian Weedbrook, Quantum Generative Adversarial Learning, Phys. Rev. Lett. 121, 040502 (2018).

ii)Vojtech Havlícek, Antonio D. Córcoles, Kristan Temme, Aram W. Harrow, Abhinav Kandala, Jerry M. Chow, and Jay M. Gambetta, Supervised learning with quantumenhanced feature spaces, Nature 567, 209 (2019)

iii)Junhua Liu, Kwan Hui Lim, Kristin L. Wood, Wei Huang,Chu Guo, and He-Liang Huang, Hybrid quantum-classicalconvolutional neural networks, Sci. China Phys. Mech.Astron. 64, 290311 (2021)

iv)Iris Cong, Soonwon Choi, and Mikhail D. Lukin, Quantum convolutional neural networks, Nat. Phys. 15, 1273 (2019).

v)Jacob Biamonte, Peter Wittek, Nicola Pancotti, Patrick Rebentrost, Nathan Wiebe, and Seth Lloyd, Quantum machine learning, Nature 549, 195 (2017).

vi) Huang, H. L., Du, Y., Gong, M., Zhao, Y., Wu, Y., Wang, C., ... & Pan, J. W. (2021). Experimental quantum generative adversarial networks for image generation. Physical Review Applied, 16(2), 024051.

Author Response

  1. This manuscript presents a quantum structure of an associative memory cell for data learning based on artificial intelligence. The manuscript is well written and clear. This result could be of interest to experts in this field. The manuscript could become suitable for publication in Sensors provided that the authors satisfactorily reply to the below comments.

The authors claim that the proposed structure could be used in machine learning in a quantum computing environment. I think some cases need to be discussed.

When the authors introduce the quantum computing in the ``Introduction'' section, it may raise interest to a broader readership if some references about the development of different quantum computing systems are provided, such as

(i) Blatt, Rainer, and Christian F. Roos. Quantum simulations with trapped ions. Nature Physics 8.4 (2012): 277-284.

(ii) Bruzewicz, Colin D., et al. Trapped-ion quantum computing: Progress and challenges. Applied Physics Reviews 6.2 (2019): 021314.

(iii) Huang, He-Liang, et al. Superconducting quantum computing: a review. Science China Information Sciences 63, 180501 (2020)

And also some references about quantum machine learning, such as

i)Seth Lloyd and Christian Weedbrook, Quantum Generative Adversarial Learning, Phys. Rev. Lett. 121, 040502 (2018).

ii)Vojtech Havlícek, Antonio D. Córcoles, Kristan Temme, Aram W. Harrow, Abhinav Kandala, Jerry M. Chow, and Jay M. Gambetta, Supervised learning with quantumenhanced feature spaces, Nature 567, 209 (2019)

iii)Junhua Liu, Kwan Hui Lim, Kristin L. Wood, Wei Huang,Chu Guo, and He-Liang Huang, Hybrid quantum-classicalconvolutional neural networks, Sci. China Phys. Mech.Astron. 64, 290311 (2021)

iv)Iris Cong, Soonwon Choi, and Mikhail D. Lukin, Quantum convolutional neural networks, Nat. Phys. 15, 1273 (2019).

v)Jacob Biamonte, Peter Wittek, Nicola Pancotti, Patrick Rebentrost, Nathan Wiebe, and Seth Lloyd, Quantum machine learning, Nature 549, 195 (2017).

  1. vi) Huang, H. L., Du, Y., Gong, M., Zhao, Y., Wu, Y., Wang, C., ... & Pan, J. W. (2021). Experimental quantum generative adversarial networks for image generation. Physical Review Applied, 16(2), 024051.

  • Thank you for your sincere advice. This study focuses on CAM design based on QCA. References [44] to [47] were mentioned in the case study, and I think the subject of the manuscript is too broad and there is not enough space to mention other quantum computing systems. We will definitely consider the reviewer's valuable advice after studying quantum machine learning.

Round 2

Reviewer 1 Report

In the previous round, it was recommended that authors must change the title but they didn't take it seriously.

Similarly, the abstract is just highlighted with yellow color and the last line has been modified but the entire body remained unchanged.

The same attitude has been adopted for contribution points and authors just highlighted with yellow color while all text remained unaltered.

At last, the conclusion is also the same except for a minor sentence revision.

My recommendations are not followed properly that was according to the standard of this journal. Anyway! if the editor thinks this manuscript is suitable then I have no further objection.

Author Response

  • I think it is the title that best expresses the subject, contribution, and purpose of the study. Based on your comments, the entire paper has been reviewed and revised again. Thank you for your help in improving the quality of the entire manuscript from abstract to conclusions. I hope for your broad understanding that I could not followed all of the your comments.

Reviewer 2 Report

The manuscript entitled: Multi-layered QCA Content-Addressable Memory Cell Using 2 Low-Power Electronic Interaction for AI-Based Data Learning 3 and Retrieval in Quantum Computing Environment has addressed all previous comments with the proper answers. However, the manuscript misses the finding and limitations of the methods and future work. There are many typo errors in the manuscript, from the introduction part to the conclusion. 

Author Response

The finding and limitations of the methods and future work are properly expressed in the analysis in Section 4 and the conclusion in Section 5. Based on your comments, the entire paper has been reviewed and revised again. Thank you for your help in improving the quality of the entire manuscript.